# Eighteenth century *Yersinia pestis* genomes reveal the long-term persistence of an historical plague focus

Kirsten I Bos[1,2]*[†], Alexander Herbig[1,2][†], Jason Sahl[3], Nicholas Waglechner[4], Mathieu Fourment[5,10,11,12], Stephen A Forrest[1], Jennifer Klunk[6,7], Verena J Schuenemann[1], Debi Poinar[6], Melanie Kuch[6], G Brian Golding[7], Olivier Dutour[8], Paul Keim[3], David M Wagner[3], Edward C Holmes[5,10,11,12], Johannes Krause[1,2]*, Hendrik N Poinar[4,6,7,9]*

[1]Department of Archeological Sciences, University of Tübingen, Tübingen, Germany; [2]Max Planck Institute for the Science of Human History, Jena, Germany; [3]Center for Microbial Genetics and Genomics, Northern Arizona University, Flagstaff, United States; [4]Michael G. DeGroote Institute for Infectious Disease Research, McMaster University, Hamilton, Canada; [5]Marie Bashir Institute for Infectious Diseases and Biosecurity, The University of Sydney, Sydney, Australia; [6]McMaster Ancient DNA Centre, Department of Anthropology, McMaster University, Hamilton, Canada; [7]Department of Biology, McMaster University, Hamilton, Canada; [8]Laboratoire d'anthropologie biologique Paul Broca, Ecole Pratique des Hautes Etudes, PACEA, Université Bordeaux, Bordeaux, France; [9]Department of Biochemistry, McMaster University, Hamilton, Canada; [10]Charles Perkins Centre, The University of Sydney, Sydney, Australia; [11]School of Life and Environmental Sciences, The University of Sydney, Sydney, Australia; [12]Sydney Medical School, The University of Sydney, Sydney, Australia

*For correspondence: bos@shh.
mpg.de (KIB); johannes.krause@
uni-tuebingen.de (JK); poinarh@
mcmaster.ca (HNP)

[†]These authors contributed equally to this work

Competing interests: The authors declare that no competing interests exist.

**Abstract** The 14th–18th century pandemic of *Yersinia pestis* caused devastating disease outbreaks in Europe for almost 400 years. The reasons for plague's persistence and abrupt disappearance in Europe are poorly understood, but could have been due to either the presence of now-extinct plague foci in Europe itself, or successive disease introductions from other locations. Here we present five *Y. pestis* genomes from one of the last European outbreaks of plague, from 1722 in Marseille, France. The lineage identified has not been found in any extant *Y. pestis* foci sampled to date, and has its ancestry in strains obtained from victims of the 14th century Black Death. These data suggest the existence of a previously uncharacterized historical plague focus that persisted for at least three centuries. We propose that this disease source may have been responsible for the many resurgences of plague in Europe following the Black Death.

## Introduction

The bacterium *Yersinia pestis* is among the most virulent pathogens known to cause disease in humans. As the agent of plague it is an existing threat to public health as the cause of both emerging and re-emerging rodent-derived epidemics in many regions of the world (*Duplantier et al., 2005*; *Vogler et al., 2011*; *Gage and Kosoy, 2005*). This, and its confirmed involvement in three major historical pandemics, have made it the subject of intense study. The first pandemic, also known as the Justinian Plague, occurred from the 6th through the 8th centuries; the second

**eLife digest** A bacterium called *Yersina pestis* is responsible for numerous human outbreaks of plague throughout history. It is carried by rats and other rodents and can spread to humans causing what we conventionally refer to as plague. The most notorious of these plague outbreaks – the Black Death – claimed millions of lives in Europe in the mid-14th century. Several other plague outbreaks emerged in Europe over the next 400 years. Then, there was a large gap before the plague re-emerged as threat in the 19th century and it continues to infect humans today, though on a smaller scale.

Scientists have extensively studied *Y. pestis* to understand its origin and how it evolved to become such a deadly threat. These studies led to the assumption that the plague outbreaks of the 14–18th centuries likely originated in rodents in Asia and spread along trade routes to other parts of the world. However, it is not clear why the plague persisted in Europe for 400 years after the Black Death. Could the bacteria have gained a foothold in local rodents instead of being reintroduced from Asia each time? If it did, why did it then disappear for such a long period from the end of the 18th century?

To help answer these questions, Bos, Herbig et al. sequenced the DNA of *Y. pestis* samples collected from the teeth of five individuals who died of plague during the last major European outbreak of plague in 1722 in Marseille, France. The DNA sequences of these bacterial samples were then compared with the DNA sequences of modern day *Y. pestis* and other historical samples of the bacteria. The results showed the bacteria in the Marseille outbreak likely evolved from the strain that caused the Black Death back in the 14th century.

The comparisons showed that the strain isolated from the teeth is not found today, and may be extinct. This suggests that a historical reservoir for plague existed somewhere, perhaps in Asia, or perhaps in Europe itself, and was able to cause outbreaks up until the 18th century.Bos, Herbig et al.'s findings may help researchers trying to control the current outbreaks of the plague in Madagascar and other places.

pandemic spanned the 14th to the 18th centuries; and the third pandemic started in the 19th century and persists to the present day.

Attempts to date the evolutionary history of *Y. pestis* using molecular clocks have been compromised by extensive variation in nucleotide substitution rates among lineages (*Cui et al., 2013*; *Wagner et al., 2014*), such that there is considerable uncertainty over how long this pathogen has caused epidemic disease in human populations. In addition, there has been lively debate as to whether or not it was the principal cause of the three historical pandemics (*Cohn, 2008*; *Scott and Duncan, 2005*). It is well established that extant lineages of *Y. pestis* circulated during the third pandemic (*Achtman et al., 2004*; *Morelli et al., 2010*), and various ancient DNA studies have now unequivocally demonstrated its involvement in the early phase of the first pandemic in the 6th century (*Harbeck et al., 2013*; *Wagner et al., 2014*) and the Black Death (1347–1351), which marks the beginning of the second plague pandemic (*Bos et al., 2011*; *Haensch et al., 2010*; *Schuenemann et al., 2011*). All three pandemics likely arose from natural rodent foci in Asia and spread along trade routes to Europe and other parts of the world (*Morelli et al., 2010*; *Wagner et al., 2014*). The strains associated with the first and second pandemics represent independent emergence events from these rodent reservoirs in Asia. The on-going third pandemic also originated in Asia, although genetic evidence suggests that it may be derived from strains that descended from those associated with the second wave that spread back to Asia and became re-established in rodent populations there (*Wagner et al., 2014*).

The impact of the second pandemic was extraordinary. During this time period there were hundreds and perhaps thousands of local plague outbreaks in human populations throughout Europe (*Schmid et al., 2015*). It is very likely that some of these outbreaks were caused by the spread of plague via the maritime transport of humans and cargo, as was undoubtedly the case during the global spread of plague during the third pandemic (*Morelli et al., 2010*). However, the processes responsible for the potential long-term persistence of plague in Europe during the second pandemic

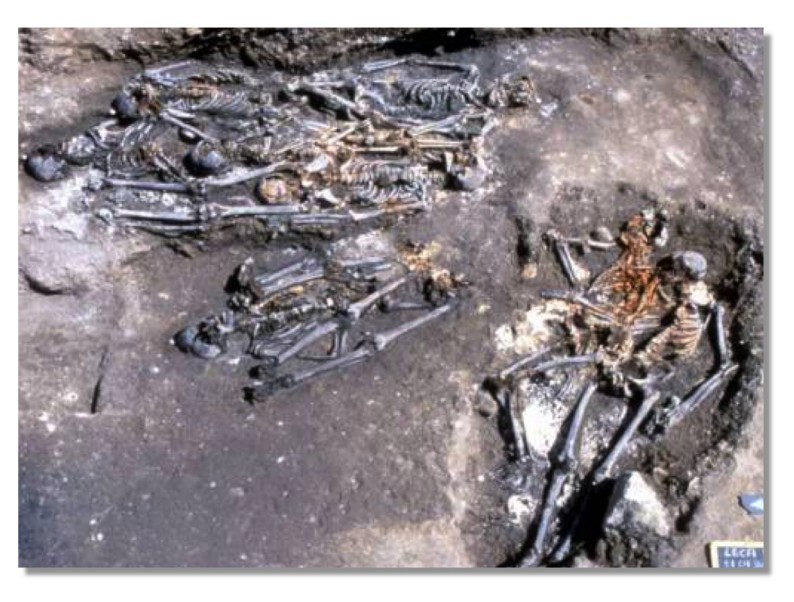

**Figure 1.** Photograph of rescue excavations at l'Observance in Marseille, France. LAPMO Université de Provence – URA 164 CNRS/AFAN.

are still subject to debate. It is possible that once introduced to Europe around the time of the Black Death, *Y. pestis* persisted there for centuries, cycling in and between rodent and human populations and being introduced or reintroduced to various regions throughout Europe (*Carmichael, 2014*). Another possibility is that plague did not persist long-term in European rodent populations, but rather was continually reintroduced from rodent plague foci in Asia (*Schmid et al., 2015*).

To address this key issue in *Y. pestis* evolution and epidemiology, we investigated plague-associated skeletal material from one of the last well-documented European epidemics, the Great Plague of Marseille (1720–1722), that occurred in the Provence region of France at the end of the second pandemic. In particular, we compared the evolutionary relationship of these historical *Y. pestis* strains to those sampled from other time periods, both modern and ancient. Our results demonstrate that the strains responsible for the Black Death left descendants that persisted for several centuries in an as yet unidentified host reservoir population, accumulated genetic variation, and eventually contributed to the Great Plague of Marseille in mid-eighteenth century France. Although these lineages no longer seem to be represented within the genetic diversity of extant sampled *Y. pestis*, they may have been involved in additional, earlier second-pandemics in the Mediterranean region and beyond.

## Results

Skeletal material used in this investigation was sampled from the Observance (OBS) collection housed in the osteoarcheological library of the Regional Department of Archaeology, French Ministry of Culture, medical faculty of Aix-Marseille Université. Historical records indicate that this collection represents a catastrophic burial for victims of the Great Plague of Marseille relapse in 1722 (*Signoli et al., 2002*). Rescue excavations carried out in 1994 unearthed the remains of 261 individuals, mostly consisting of adult males (*Figure 1*). Material from this collection was also used in the first PCR-based investigation of ancient *Y. pestis* DNA from archaeological tissue (*Drancourt et al., 1998*). In our study 20 in situ teeth were sampled and screened for *Y. pestis* DNA through a quantitative PCR assay. Amplification products were detectable for five of the 20 teeth (*Table 1*). Products were not sequenced. Negative controls were free of amplification products. The five putatively *Y. pestis*-positive samples were subject to high-

**Table 1.** Quantitative PCR data for the *pla* gene obtained for the 19 Observance teeth extracted. Values below std curve detection are highlighted in grey.

| PCR Blks: | Copies per µl |
| --- | --- |
| PCR Blk 1 | 0 |
| PCR Blk 2 | 0 |
| PCR Blk 3 | 0 |
| PCR Blk 4 | 0 |

| Extraction Blks: | Copies per µl |
| --- | --- |
| BLk 1 | 0 |
| Blk 2 | 0 |
| Blk 3 | 0 |
| Blk 4 | 0 |

*Extracts:*

| Sample | Copies per µl |
| --- | --- |
| 101a | 0 |
| 101b | 0 |
| 102 | 0 |
| 103 | 0 |
| 107 | 39 |
| 108 | 0 |
| 109 | 0 |
| 110 | 50 |
| 116 | 105 |
| 117a | 0 |
| 117b | 1 |
| 118 | 0 |
| 123 | 0 |
| 124 | 142 |
| 125 | 0 |
| 126 | 0 |
| 130 | 0 |
| 134 | 1 |
| 137 | 30 |
| 144 | 0 |

throughput DNA sequencing after array-based enrichment for *Y. pestis* DNA. Mapping to the *Y. pestis* CO92 reference genome revealed a minimum of 12-fold average genomic coverage for each of the five genomes (*Table 2* and *Figure 2*). Raw sequencing data as well as mapped reads have been deposited at the European Nucleotide Archive under accession PRJEB12163. Comparative analysis together with 133 previously published *Y. pestis* genomes resulted in 3109 core genome SNPs for phylogenetic analysis. No homoplasies were identified.

The phylogenetic position of the five OBS sequences on a unique branch of the *Y. pestis* phylogeny (lineages shown in red in *Figure 3*), yet seemingly derived from those strains associated with the Black Death, both confirms its authenticity and reveals that it is likely a direct descendent of strains that were present in Europe during the Black Death. Although the topological uncertainty (i.e. trichotomy) among the second pandemic strains (London and Observance) suggests that there was a radiation of *Y. pestis* lineages at the time, it is notable that the OBS sequences do not possess the additional derived position that is present in both the 6330 London strain and the SNP-typed individual from Bergen Op Zoom (defined as the 's12' SNP by *Haensch et al., 2010*). Indeed, the tree is striking in that there is clear phylogenetic support (88% bootstrap values, *Figure 3A*) for 6330 clustering with the Branch 1 strains, including those that gave rise to the third (modern) plague pandemic. Finally, although evolutionary rates in *Y. pestis* are notoriously variable (*Cui et al., 2013*; *Wagner et al., 2014*), the comparatively late time period from which our samples derive and the long branch leading to these OBS sequences suggests that this lineage co-existed for an extended period with the Branch 1 strains responsible for all later human plague outbreaks outside of China.

## Identification of DFR4 in OBS strains

Using the pan-array design, we were able to identify a ~15 kb Genomic Island (GI) in the OBS strains as previously observed in the Justinian and Black Death strains (*Wagner et al., 2014*) and referred to as DFR 4 (difference region 4). Notably, this region has been lost in some *Y. pestis* strains such as CO92. This island is a striking example of the decay common to the highly plastic *Y. pestis* genome.

**Table 2.** Mapping statistics.

| Sample ID | Raw read pairs | Preprocessed reads | Mapped reads | % mapped | Dedupped | Duplication factor | Fold coverage | % covered |
|---|---|---|---|---|---|---|---|---|
| OBS107 | 14,646,710 | 15,395,287 | 1,850,734 | 12.02% | 851,096 | 2.17 | 12.20 | 84.79 |
| OBS110 | 9,879,257 | 10,458,571 | 1,581,158 | 15.12% | 855,988 | 1.85 | 14.61 | 90.63 |
| OBS116 | 16,858,273 | 17,876,883 | 2,931,554 | 16.40% | 1,363,415 | 2.15 | 20.04 | 92.56 |
| OBS124 | 48,797,602 | 52,345,884 | 4,972,347 | 9.50% | 973,763 | 5.11 | 13.34 | 85.37 |
| OBS137 | 120,018,142 | 129,095,434 | 12,669,625 | 9.81% | 1,583,596 | 8.00 | 24.40 | 92.76 |

## Discussion

The Observance lineage of *Y. pestis* identified by our analysis of material from the Great Plague of Marseille (1720–1722) is clearly phylogenetically distinct and has not been identified in any extant plague foci for which full genome data are available. That this lineage is seemingly confined to these ancient samples suggests that it is now extinct, or has yet to be sampled from extant reservoirs. Based on available data, the *Y. pestis* strains most closely related to the Observance lineage are those obtained from other ancient DNA studies of victims of the Black Death in London and other areas (*Haensch et al., 2010*; *Bos et al., 2011*) that occurred some 350 years earlier.

The most important conclusion from the current study is that *Y. pestis* persisted and diversified in at least two lineages throughout the second pandemic, one represented by our Observance strain, and the other by London St. Mary Graces individual 6330. The absence of currently sequenced modern *Y. pestis* genomes that share an immediate phylogenetic relationship with the Observance lineage suggests that it may no longer exist. The length of the branch leading to the Observance lineage is consistent with a long history of circulation. Of note, this new lineage does not share the single derived mutation that is common to the London St. Mary Graces genome and the Bergen op Zoom plague victim from the Netherlands (*Haensch et al., 2010*); hence, multiple variants of *Y. pestis* evidently circulated in London and elsewhere in the 14th century, and at least one of these variants followed a path that differs from the lineage that later gave rise to the 18th century plague epidemic in Provence.

Since the novel lineage of *Y. pestis* identified here was obtained from an active Mediterranean port city that served as a main commercial hub and entry point into Western Europe from various origins, the precise location of the disease's source population cannot be easily determined. Several scenarios can explain these data. First, all *Y. pestis* diversity documented in the Black Death and later plague outbreaks could have stemmed from foci in Asia, where European epidemics were the result of successive introductions from this distant, albeit prolific, source population (*Schmid et al., 2015*). This model would not require the back migration of plague into Asia for the third pandemic proposed elsewhere (*Wagner et al, 2014*), since all preexisting diversity would already be present. A multiple wave theory has previously been proposed to explain the presence of a reportedly different 14th century lineage in the Low Countries (defined by their single 's12' SNP) compared to those circulating in England and France (*Haensch et al., 2010*). However, the presence of this branch 1 position in both the ancestral and subsequently the derived state in two temporally close London outbreaks (*Bos et al., 2011*; *Figure 3B*) suggests that it became fixed during the Black Death or its immediate aftermath, rather than having been introduced via a separate pulse from an Asian focus. To date, all of the *Y. pestis* material examined from the second pandemic has been assigned to Branch 1 in the *Y. pestis* phylogeny, indicating a close evolutionary relationship. As the large "'big bang' radiation of *Y. pestis* at the beginning of the second pandemic (*Cui et al., 2013*) established at least four new lineages, it seems unlikely that independent waves of plague from Asia into Europe during the Black Death would have involved strains that group on the same branch and differ by so few positions. As China harbours many branch 1 descendent lineages that also share this 's12'" SNP, our observation adds weight to the notion that the branch 1 lineage of *Y. pestis* spread east sometime during the second pandemic after it entered Europe, and subsequently became established in one or more East Asian reservoir species (*Wagner et al., 2014*). Here it remained before radiating

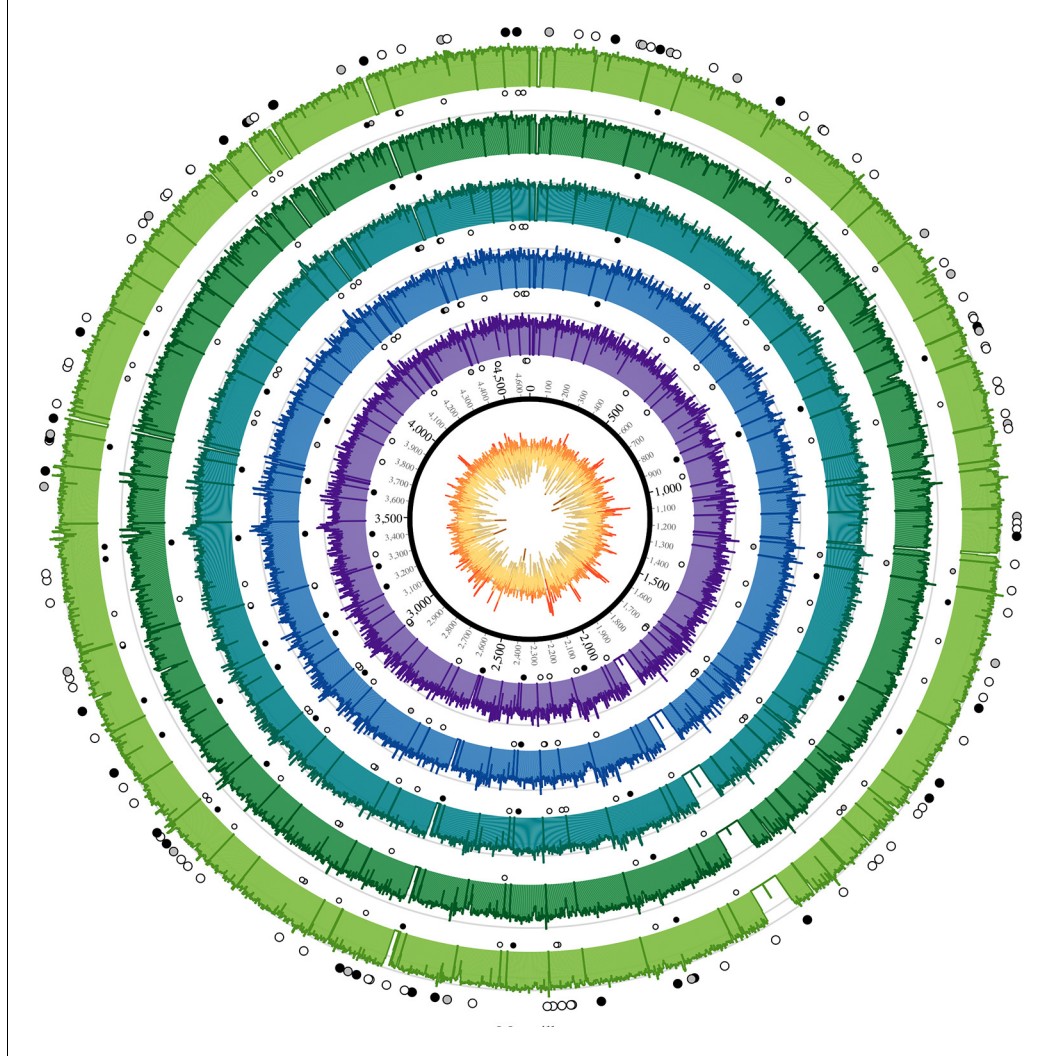

**Figure 2.** Coverage plots for all reconstructed Observance core genomes. Inner ring: GC content from low (brown, <30%) to high (orange, >55%). Outer concentric rings: from inner to outer, coverage plots of OBS107, 110, 116, 124, and 137 on a logarithmic scale. Axes are at 30x and 100x. Dots: SNPs (black = non-synonymous, grey = synonymous, white = intergenic). Outer ring (larger dots) is SNPs shared by all five strains. Inner rings (smaller dots) are associated with the strain immediately outside and are SNPs that are not shared by all five strains (but may be shared by 2–4 strains). Krzywinski, M. *et al.* Circos: an Information Aesthetic for Comparative Genomics. Genome Res (2009) 19:1639–1645

to other locations in the late 19[th] century, giving rise to the current worldwide 'third wave' *Y. pestis* pandemic.

In our view, a far more plausible option to account for the distinct position of the Observance lineage is that a *Y. pestis* strain responsible for the Black Death became established in a natural host reservoir population within Europe or western Asia. Once established, this *Y. pestis* lineage evolved locally for hundreds of years and contributed to repeated human epidemics in Europe. A similar scenario is possible for the lineage(s) identified in the London St. Mary Graces and Bergen op Zoom material: like Observance, both descend from the Black Death and together they represent at least one source of European plague distinct from that which later caused the Great Plague of Marseille. Plague's temporary persistence somewhere on the European mainland is a compelling possibility (*Carmichael, 2014*). Alternatively, reservoir populations located in geographically adjacent regions, such as in western Asia or the Caucasus, may have acted as regular sources of disease through successive westerly pulses during the three century-long second pandemic (*Schmid et al., 2015*). Since our study used material exclusively from a highly active port with trade connections to

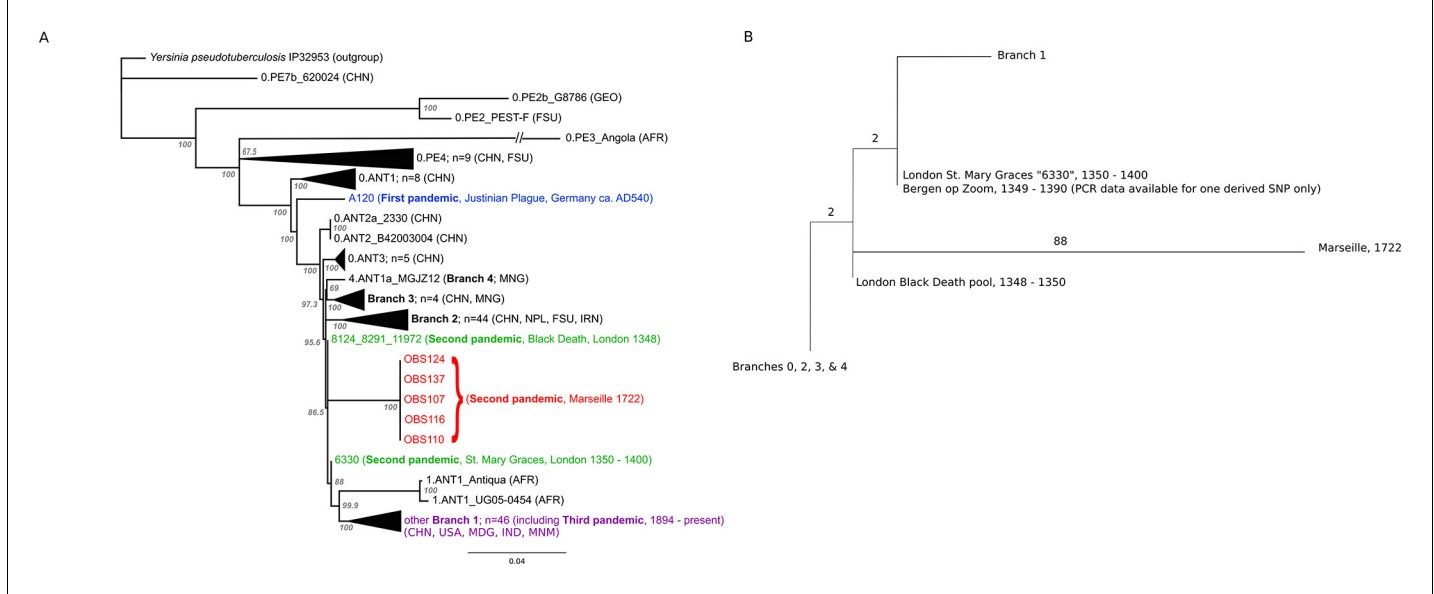

**Figure 3.** Phylogenetic tree for *Yersinia pestis*. (A) Maximum likelihood phylogeny of *Y. pestis* genomic SNPs showing the position of the Observance (OBS) lineage (red) relative to those of 130 modern (black) and three ancient strains (Black Death in green and Justinian Plague in blue). Modern strains from the third pandemic are shown in purple to highlight their close genetic relatedness. Monophyletic groups of sequences have been collapsed to improve clarity and are shown as triangles. The tree is rooted using single strain of *Y. pseudotuberculosis* (IP32953), with all derived SNPs removed to assist scaling, with branch lengths reflecting the number of nucleotide substitutions/SNP site. The length of the branch leading to the 0.PE3_Angola (AFR) lineage was reduced because its excessive length adversely affected the scaling of the tree. Location abbreviations are as follows: CNH (China), GEO (Georgia), FSU (Former Soviet Union), MNG (Mongolia), NPL (Nepal), IRN (Iran), AFR (Africa), USA (United States of America), MDG (Madagascar), IND (India). (B) Expanded phylogeny schematic to show the relative positions of the Black Death and the Observance lineages. Numbers on branches correspond to SNPs.

many areas in the Mediterranean and beyond, it is impossible to identify a source population for the Great Plague of Marseille at our current resolution.

Our analysis reveals a previously uncharacterized plague source that could have fuelled the European epidemics from the time of the Black Death until the mid-eighteenth century. This parallels aspects of the third pandemic, that rapidly spread globally and established novel, long-lived endemic rodent foci that periodically emerge to cause human disease (*Morelli et al., 2010*). The current resolution of *Y. pestis* phylogeography suggests that the historical reservoir identified here may no longer exist. More extensive sampling of both modern rodent populations and ancient human, as well as rodent, skeletal remains from various regions of Asia, the Caucasus, and Europe may reveal additional clues regarding past ecological niches for plague. The reasons for the apparent disappearance of this historic natural plague focus are, however, unknown, and any single model proposed for plague's disappearance is likely to be overly simplistic. Pre-industrial Europe shared many of the same conditions that are correlated with plague dissemination today including pronounced social inequality, and poor sanitation coupled with high population density in urban centers (*Vogler et al., 2011*; *Barnes, 2014*). This constellation of anthropogenic factors, along with the significant social and environmental changes that occurred during the Industrial Revolution must be considered alongside models of climate and vector-driven dynamics as contributing to the rapid decline in historical European plague outbreaks, where genomic data provide but one piece of critical information in this consilient approach.

## Material and methods

For this study 20 in situ teeth were freshly harvested, horizontally sectioned at the cementoenamel junction in a dedicated ancient DNA facility, and drilled to remove approximately 50 mg of dental pulp/dentin from the inner surface of the tooth crown or the roots. DNA extractions were carried out following protocols described elsewhere (*Schwarz et al., 2009*; *Rohland and Hofreiter, 2007*;

*Dabney et al., 2013*), and survival of *Y. pestis* DNA was evaluated through an established quantitative PCR assay for the *pla* gene (*Schuenemann et al., 2011*). DNA extracts for the five putatively *Y. pestis*-positive samples, each from a separate individual, were converted into double-indexed Illumina libraries (*Meyer and Kircher, 2010*; *Kircher et al., 2012*) following an initial uracil DNA glycosylase treatment step (*Briggs et al., 2010*) to remove deaminated cytosines, the most common form of ancient DNA damage. Libraries were amplified with AccuPrime Pfx (Life Technologies, Germany) to a total of 20 µg and serially captured in equimolar concentrations over two identical 1 million-feature Agilent (Santa Clara, CA) microarrays (*Hodges et al., 2009*). Arrays were designed with probes tiled every three base pairs matching the CO92 *Y. pestis* reference chromosome (NC_003143), supplemented with additional chromosomal regions from *Y. pestis* biovar Microtus str. 91,001 (NC_005810), *Y. pseudotuberculosis* IP 32,953 (NC_006155) and *Y. pseudotuberculosis* IP 31,758 (NC_009708) that are absent in CO92. Captured products, including those from negative controls, were sequenced on one lane of an Illumina HiSeq 2000.

Raw reads were trimmed, overlapping paired reads were merged as described elsewhere (*Schuenemann et al., 2013*), and merged reads were subsequently filtered for a minimal length of 30 bp. Preprocessed reads were mapped to the CO92 reference genome using the Burrows-Wheeler Aligner (BWA) (*Li and Durbin, 2010*) with increased specificity (-n 0.1) and a map quality filter of 37.

For maximum accuracy in SNP calling, reads were processed independently at three research centres, and the intersection was used as our final SNP table (*Supplementary file 1*). At the University of Tuebingen, reads were processed as described above and SNP calling was done according to a protocol described by *Bos et al. (2014)* using the UnifiedGenotyper of the Genome Analysis Toolkit (GATK) (*DePristo et al., 2011*). Data for 133 previously published *Y. pestis* strains were processed respectively (*Supplementary file 2*). A custom tool was used for the comparative processing of the results. SNPs were called with a minimal coverage of 5-fold and a minimal frequency of 90% for the SNP allele. Problematic sites as identified by *Morelli et al. (2010)*, genomic non-core regions as defined by *Cui et al. (2013)* as well as annotated repeat regions, rRNAs, tRNAs and tmRNAs were excluded from the SNP analysis. SNPs were annotated using SnpEff (*Cingolani et al., 2012*) with default parameters. The upstream and downstream region size was set to 100 nt. At McMaster University, raw reads were trimmed and merged using SeqPrep (https://github.com/jstjohn/SeqPrep), requiring a minimum overlap of 11 base pairs to merge. Reads shorter than 24 base pairs were filtered out using SeqPrep. Merged reads and unmerged forward reads were concatenated and mapped to the CO92 reference sequence (NC_003143) using the Burrows-Wheeler Aligner (BWA) version 0.7.5 (Li and Durbin, 2009) with the parameters described in *Wagner et al. (2014)*. Duplicates were collapsed using a custom script, which collapses only reads with identical 5' position, 3' position, and direction. Assemblies were imported into Geneious R6 and SNPs with at least 5-fold coverage and 90% variance were called. SNPs were visually inspected for quality. At NAU, reads were trimmed with Trimmomatic (*Bolger et al., 2014*) and aligned against the CO92 reference sequence using BWA-MEM. SNPs were called with the Unified Genotyper in GATK and were filtered by minimum coverage (5X) and allele frequency (90%). SNPs in genome assemblies were identified by a direct mapping against the reference sequence using NUCmer (*Delcher et al., 2002*). These methods were wrapped by the Northern Arizona SNP Pipeline (NASP) (tgennorth.github.io/NASP).

## Phylogenetic analysis of the observance *Y. pestis*

A phylogenetic tree of *Y. pestis* strains was inferred using the maximum likelihood (ML) method available in PhyML (*Guindon et al., 2010*), assuming the GTR model of nucleotide substitution (parameter values available from the authors on request) and a combination of NNI and SPR branch-swapping. The robustness of individual nodes was assessed using bootstrap resampling (1000 pseudo-replicates) using the same substitution model and branch-swapping procedure as described above. The phylogeny comprised data from the five Observance (OBS) strains, one sequence from Black Death victims who died in London between 1348 and 1350 (a combined pool of identical strains 8291, 11972, and 8124), one sequence from a subsequent outbreak in London, 1350 – 1400 (individual 6330), one sequence from the Plague of Justinian, AD540 (strain A120), and 130 modern *Y. pestis* strains. A single sequence of *Y. pseudotuberculosis* (strain IP32953) was used as an outgroup to root the tree, although with all derived SNPs removed to assist branch-length scaling.

## Identification of DFR4 in OBS strains

Merged and trimmed reads were mapped to the *Y. pestis* biovar Microtus str. 91,001 chromosome at the DFR 4 region (between position 1041000 and 1063000). Additionally, the unique Microtus 31-mers were also mapped back to the Microtus chromosome to ensure identity of the region using short read length data. All resulting alignment files were compared using BEDTools (*Quinlan and Hall, 2010*) to identify intervals of the Microtus chromosome that were unique to Microtus when compared to CO92 and were covered by reads from our ancient extracts.

## Acknowledgements

We acknowledge The Director of the Regional Department of Archeology (DRAC-PACA / Aix-en-Provence, France) for granting us access to the Observance skeletal collection (4422-17/09/2009). Funding was provided by European Research Council starting grant APGREID (to JK, KIB and AH), Social Sciences and Humanities Research Council of Canada postdoctoral fellowship grant 756-2011-501 (to KIB), National Health and Medical Research Council (NHMRC) grant (to ECH and HNP), and an NHMRC Australia Fellowship to ECH, as well as a Canada Research Chair and NSERC discovery research grant to HNP. We thank M Green and SN DeWitte for providing us with the corrected date for 6330. HNP thanks former and current members of the McMaster aDNA Centre for their help with this study.

## Additional information

### Funding

| Funder | Author |
|---|---|
| Social Sciences and Humanities Research Council of Canada | Kirsten I Bos |
| National Health and Medical Research Council | Edward C Holmes<br>Hendrik N Poinar |
| European Research Council | Johannes Krause |
| Natural Sciences and Engineering Research Council of Canada | Hendrik N Poinar |

The funders had no role in study design, data collection and interpretation, or the decision to submit the work for publication.

### Author contributions

KIB, Conception and design, Acquisition of data, Analysis and interpretation of data, Drafting or revising the article; AH, JS, NW, MF, JKl, GBG, DMW, Analysis and interpretation of data, Drafting or revising the article; SAF, PK, ECH, JKr, HNP, Conception and design, Analysis and interpretation of data, Drafting or revising the article; VJS, MK, Acquisition of data, Drafting or revising the article; DP, Conception and design, Drafting or revising the article; OD, Conception and design, Acquisition of data, Drafting or revising the article

### Author ORCIDs

Johannes Krause, http://orcid.org/0000-0001-9144-3920

## Additional files

### Supplementary files

• Supplementary file 1. Combined SNP table from three SNP calling approaches.

• Supplementary file 2. List of genomes used in SNP calling and phylogeny.

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
