## [Decision Letter]

Thank you for submitting your work entitled "Eighteenth century *Yersinia pestis* genomes reveal the long-term persistence of an historical plague focus" for consideration by *eLife*. Your article has been reviewed by three peer reviewers, and the evaluation has been overseen by a Reviewing Editor and Richard Losick as the Senior Editor.

The reviewers have discussed the reviews with one another and the Reviewing Editor has drafted this decision to help you prepare a revised submission.

Summary:

All reviewers stressed the importance of ancient genomes to unravel the mysteries of historical epidemics and appreciated the technological difficulties involved. The new genomes belong to a previously unsampled branch of the *Y. pestis* phylogeny that has its origin in the radiation at the time of the Black Death. This is an important finding.

Essential revisions:

We did not feel that the five almost identical genomes and their phylogenetic relationship to other genomes provide enough evidence to justify the focus on one particular hypothesis. Instead, the manuscript should discuss the evidence for different scenarios in a much more balanced way and explicitly address assumptions and weaknesses of the different hypotheses.

There seem to be two major competing hypotheses (with somewhat fuzzy distinction):

1) All lineages come out of Asia and diversity in 14th century Europe was imported from preexisting diversity in Asia. This scenario would not require the back-migration of branch 1 to Asia. The OBS strains could be reintroduced from Asia or have persisted somewhere else, we can't tell. You argue that it is improbable that multiple so similar lineages were imported independently to Europe from Asia. However, the majority of modern lineages descend directly from the "big bang" radiation (either slightly ancestral to the London genomes or via branch 1) and there is no reason why the OBS lineage should behave differently. Evidently the "big bang" wiped out much of *Y. pestis* diversity around 1300.

2) Alternatively a single strain was introduced into Europe, diversified slightly and remigrated to Asia. The OBS lineages branched off, persisted somewhere in the Mediterranean (or beyond) and caused the Marseille outbreak. The argument against reintroduction from Asia – the lineage sharing 2 derived mutations (bootstrap 86%) with the other ancient European isolates – is weak since we don't know where these derived mutations arose (preexisting in Asia versus Europe/in transit).

In the light of this ambiguity, the title, Abstract and Discussion should be revised extensively. A tabular or visual summary of the different hypotheses and the arguments in favour of or against the scenarios would be useful.

---

## [Author Response]

Essential revisions:

*We did not feel that the five almost identical genomes and their phylogenetic relationship to other genomes provide enough evidence to justify the focus on one particular hypothesis. Instead, the manuscript should discuss the evidence for different scenarios in a much more balanced way and explicitly address assumptions and weaknesses of the different hypotheses. There seem to be two major competing hypotheses (with somewhat fuzzy distinction):*

*1) All lineages come out of Asia and diversity in 14th century Europe was imported from preexisting diversity in Asia. This scenario would not require the back-migration of branch 1 to Asia. The OBS strains could be reintroduced from Asia or have persisted somewhere else, we can't tell. You argue that it is improbable that multiple so similar lineages were imported independently to Europe from Asia. However, the majority of modern lineages descend directly from the "big bang" radiation (either slightly ancestral to the London genomes or via branch 1) and there is no reason why the OBS lineage should behave differently. Evidently the "big bang" wiped out much of Y. pestis diversity around 1300. 2) Alternatively a single strain was introduced into Europe, diversified slightly and remigrated to Asia. The OBS lineages branched off, persisted somewhere in the Mediterranean (or beyond) and caused the Marseille outbreak. The argument against reintroduction from Asia – the lineage sharing 2 derived mutations (bootstrap 86%) with the other ancient European isolates – is weak since we don't know where these derived mutations arose (preexisting in Asia versus Europe/in transit). In the light of this ambiguity, the title, Abstract and Discussion should be revised extensively. A tabular or visual summary of the different hypotheses and the arguments in favour of or against the scenarios would be useful.*

We thank the reviewers for their helpful consolidated comments, and feel they have clearly improved the manuscript. We very much want to avoid conclusions that overstate our data. In light of this, we have restructured and edited the Introduction and Discussion sections of the paper to better describe the different scenarios compatible with the data we present here. However, we do feel that the scenario under which all *Y. pestis* diversity stems from China, including the OBS lineage, is the least plausible and is not well supported by the existing data. While we’ve entertained this possibility in greater detail, we more comprehensively explain its weaknesses. Indeed, because the OBS strain falls so close to the Black Death strains it seems highly unlikely to us that they would both be independent importations from East Asia. In particular, why would the re-imported strain (i.e. OBS) be this lineage rather than the myriad of others in Asia? As for the alternative scenarios of a European source population or one from an adjacent area such as West Asia or the Caucasus, we have no data to clearly favour one over the other. Hence we present them both with equal likelihood in the revised version of the paper. We have opted not to change the title of the manuscript, since it makes no reference to possible interpretations of the data. Regarding the suggestion of a tabular or visual depiction of the various scenarios, there are simply too many unknowns for this to be viable; our attempts at simplifying them have seemed inadequate and are potentially misleading and overly complex. For this reason, we have opted to present them in text only. We sincerely hope the reviewers understand our rationale here and, overall, we believe that we have revised the paper according to the reviewers’ wishes.